# Relationships Between Cumulative Family Risk, Emotion Regulation Difficulties, and Non-Suicidal Self-Injury in Adolescents: A Person-Centered Analysis

**DOI:** 10.3390/bs15040543

**Published:** 2025-04-17

**Authors:** Xiaoxian Liu, Hengyuan Fan, Ruijuan Xiong, Lei An, Yiming Wang, Ruojuan Du, Xiaosheng Ding

**Affiliations:** Faculty of Education, Henan Normal University, Xinxiang 453007, China; 111028@htu.edu.cn (X.L.); 2310283157@stu.htu.edu.cn (H.F.); 2110483175@stu.htu.edu.cn (R.X.); 2023249@htu.edu.cn (L.A.); 2410283165@stu.htu.edu.cn (Y.W.); 2410283163@stu.htu.edu.cn (R.D.)

**Keywords:** cumulative family risk, emotion regulation difficulties, non-suicidal self-injury, latent class analysis, latent profile analysis, adolescent

## Abstract

The present study used a person-centered approach to examine the latent patterns of cumulative family risk and emotion regulation difficulties in adolescents and their relationships with non-suicidal self-injury. A sample of 1046 primary and secondary school students was analyzed using latent class analysis and latent profile analysis to identify subgroups of cumulative family risk and emotion regulation difficulties, respectively. The results were as follows: (1) Two latent classes of cumulative family risk were identified: a high-risk group (30.78%) and a low-risk group (69.22%). Adolescents in the high-risk group had significantly higher self-injury scores. (2) Three latent profiles of emotion regulation difficulties were identified: a low-difficulty group (56.02%), a medium-difficulty group (32.60%), and a high-difficulty group (11.38%). Adolescents in the high-difficulty group had the highest self-injury scores. (3) The logistic regression showed that adolescents in the high-risk group were more likely to belong to the high-difficulty group, followed by the medium- and low-difficulty groups. In summary, adolescents with high levels of cumulative family risk are also more likely to exhibit high levels of emotion regulation difficulties and self-injurious behavior.

## 1. Introduction

Non-suicidal self-injury (NSSI) refers to the deliberate, self-inflicted damage to body tissue without suicidal intent and for reasons not socially accepted, typically involving behaviors such as cutting or burning ([26]). NSSI is a significant mental health concern that negatively impacts the physical and psychological well-being of adolescents ([44]) and is associated with an increased risk of suicidal ideation and behaviors ([21]).

The Systemic Model of Self-Injury suggests that adolescent NSSI arises from dysfunction in family or social environments ([6]). Risk factors within the family environment, such as disrupted family structure, limited resources, and a lack of parent–child closeness, may impair the development of children’s emotion regulation abilities ([11]). According to the Affective Cascade Model, emotional dysregulation and the accumulation of negative emotions are critical to the initiation and development of NSSI ([34]).

Numerous studies have examined the relationship between family factors and adolescent NSSI, with a particular focus on the influence of cumulative family risk ([31]; [39]; [23]). Previous studies have mostly relied on a variable-centered approach (e.g., regression analysis), which may overlook adolescents’ individual differences. Furthermore, studies on the relationship between family risk and NSSI in China have primarily focused on junior and senior high school students. However, evidence from international studies shows a high prevalence of NSSI in young adolescents, ranging from 7.6% to 23.7% ([1]; [38]). Early adolescence is characterized by rapid emotional, cognitive, and social transitions, and is recognized as a critical developmental period for the emergence of maladaptive behaviors, such as NSSI ([35]). Identifying the factors associated with NSSI during this stage is essential for assessing both current and future risks and has important implications for prevention and early intervention efforts ([7]). Given these limitations, we used a person-centered approach to examine the relationships between cumulative family risk, emotion regulation difficulties, and NSSI during early adolescence.

### 1.1. Cumulative Family Risk and Adolescent NSSI

Family risk encompasses factors within the family microsystem that may increase the likelihood of negative developmental outcomes ([3]). It typically includes three components ([42]): (1) structure risk, such as parental divorce or parent–child separation; (2) resource risk, involving deficits in economic, educational, or material resource; and (3) family atmosphere risk, characterized by a lack of emotional support or parent–child closeness. [48]’ ([48]) Developmental Psychopathology Model of Self-Injury suggests that family risk hinders the development of essential psychological capacities (e.g., emotional regulation ability), with NSSI then serving as a compensatory mechanism to manage psychological challenges. Numerous studies have shown that family risk factors, such as left-behind experiences, low parental education, disrupted family structure, family economic pressure, family conflict, and low parent–child closeness have been shown to be predictors of adolescent NSSI ([46]; [23]; [42]).

The Cumulative Risk Effect Theory suggests that while a single risk factor may not significantly impact individual development, the likelihood of adverse outcomes increases considerably when multiple risks are experienced simultaneously ([9]). The accumulation of multiple family risk factors may be a more accurate predictor of adolescent NSSI than a single risk factor ([4]). In the current study, we used the person-centered approach of exploratory latent class analysis (LCA) to identify latent patterns of cumulative family risk among adolescents and to examine differences in NSSI across these classes.

### 1.2. Emotion Regulation Difficulties and Adolescent NSSI

Emotion regulation difficulties involve challenges in identifying, understanding, accepting, and managing emotions ([13]). The Integrated Model of Self-Injury highlights emotion regulation as a key internal factor that predicts NSSI ([25]). The Affective Cascade Model posits that emotional dysregulation and the accumulation of negative emotions play significant roles in the initiation and maintenance of NSSI ([34]). Compared to non-injurers, individuals who engage in NSSI have been shown to experience greater challenges in regulating negative emotions ([45]; [24]; [8]).

According to the Multidimensional Model of Emotion Regulation Difficulties, there are six dimensions of emotion regulation: emotional response acceptance, goal-directed behavior, impulse control, emotional awareness, emotional clarity, and use of regulation strategies ([13]). Individual differences in these dimensions may form distinct latent patterns of emotion regulation difficulties ([5]), each potentially linked to different associations with NSSI ([30]). In the current study, we used the latent profile analysis (LPA), which is a variation in the latent class analysis used when the observed variables are continuous ([43]), to identify latent patterns of emotion regulation difficulties among adolescents and to examine differences in NSSI across these profiles.

### 1.3. Cumulative Family Risk and Emotion Regulation Difficulties

Evidence indicates that family risk significantly predicts children’s emotional insecurity and their use of emotion regulation strategies ([15]), and emotion regulation difficulties may be more serious in children exposed to cumulative family risk factors ([12]). Therefore, further exploration is needed to understand how different patterns of cumulative family risk affect adolescents’ emotion regulation difficulties. We used logistic regression to further examine the predictive role of latent classes of cumulative family risk on latent profiles of emotion regulation difficulty among adolescents.

### 1.4. The Present Study

First, we conducted LCA to identify latent patterns of cumulative family risk among adolescents and examined differences in NSSI across the identified classes. Second, LPA was conducted to identify latent patterns of emotion regulation difficulties in adolescents, followed by an analysis of the differences in NSSI across the identified profiles. Finally, logistic regression was used to assess latent classes of cumulative family risk as predictors of latent profiles of emotion regulation difficulty in adolescents.

The following hypotheses were proposed: (1) adolescents’ cumulative family risk can be categorized into distinct latent classes; (2) adolescents’ emotion regulation difficulties can be categorized into distinct latent profiles; (3) adolescents with higher levels of cumulative family risk will report higher levels of NSSI; (4) adolescents with higher levels of emotion regulation difficulties will report higher levels of NSSI; (5) adolescents in the higher-risk family group will be more likely to belong to latent profiles characterized by greater emotion regulation difficulties.

Person-centered analysis (e.g., LCA and LPA) identifies subgroups within a population that share similar characteristics. In contrast to variable-centered analysis, which examines average effects across all individuals, this method emphasizes individual heterogeneity ([2]). This approach not only refines the precision of variable relationships but also offers insights for designing targeted interventions. Consequently, researchers can customize interventions to address the distinct needs of various subgroups effectively ([43]).

## 2. Materials and Methods

### 2.1. Participants

The sample included students from the 5th and 6th grade of an elementary school and from the 7th and 8th grade of a middle school in a city in Henan Province, China. A total of 1173 questionnaires were distributed using cluster sampling. Finally, 1046 valid responses were retained, yielding a response rate of 89.2%, comprising 330 students in the 5th grade (28.8%), 257 in the 6th grade (22.4%), 239 in the 7th grade (20.9%), and 220 students in the 8th grade (19.2%). Of these, 528 were boys (50.5%) and 518 were girls (49.5%).

### 2.2. Procedure

Before the data collection, the researchers communicated thoroughly with the school and obtained informed consent from school officials, parents, and students. The assessment was administered in class groups, with trained psychology graduate students serving as examiners. The assessment lasted 25 min, after which the examiners collected the questionnaires.

Given the sensitive nature of NSSI, several measures were implemented to minimize potential distress among participants. Prior to the commencement of the research, individuals were informed that the questionnaire would include items related to self-injury and emotional experiences. Participation was fully voluntary, and participants retained the right to withdraw at any point without consequence. If any participant experienced discomfort during the survey, the assessment was immediately discontinued, and psychological support was made available, including individual counseling and group-based interventions.

### 2.3. Measures

#### 2.3.1. Adolescent Self-Injury Questionnaire

The Revised Adolescent Self-Injury Questionnaire ([10]), comprising 18 multiple-choice items. Self-injury was assessed by calculating the product of its frequency and severity, with higher scores reflecting higher levels of self-injury. The frequency is assessed using a 4-point scale (0 = never, 3 = often), while the severity is assessed using a 5-point scale (0 = no harm, 4 = very severe harm). A score of 1 or greater over the past year was used as the threshold for the presence of self-injury. In the present study, the construct validity of the scale was supported by confirmatory factor analysis (CFA), indicating an acceptable fit to the data (χ^2^(135) = 388.86, CFI = 0.93, TLI = 0.91, RMSEA = 0.06, SRMR = 0.05). The internal consistency reliability was satisfactory, with a Cronbach’s α of 0.81.

#### 2.3.2. Cumulative Family Risk Questionnaire

The cumulative family risk was assessed in terms of six factors. Each factor was coded as present or absent (0 = no risk, 1 = at risk), and the cumulative family risk index was calculated by summing the risk scores ([4]).

(1) Left-behind experience: Codes as 1 if the participant reported that one or both parents had been away for work for more than six months; otherwise, it was coded as 0 ([42]).

(2) Parental education level: Coded as 1 if the participant reported that both parents’ education levels were below high school; otherwise, it was coded as 0 ([42]).

(3) Family structure: Coded as 1 if the participant reported they did not live with both their biological father and mother; otherwise, it was coded as 0 ([46]).

(4) Family economic pressure: Assessed using the Family Economic Pressure Scale ([40]). Participants scoring in the top 25% were coded as 1, while others were coded as 0. In this study, the scale demonstrated a Cronbach’s α of 0.79.

(5) Parent–child closeness: Assessed using the Parent–Child Closeness Scale ([49]). Participants scoring in the bottom 25% were coded as 1, while others were coded as 0. The father–child and mother–child subscales had Cronbach’s α of 0.87 and 0.85.

(6) Family conflict: Assessed using the Conflict Subscale of the Family Environment Scale ([36]). Participants scoring in the top 25% were coded as 1, while others were coded as 0. In this study, the scale demonstrated a Cronbach’s α of 0.65.

In line with previous research practices, risk factors were selected according to the following principles ([47]): (1) relevance: risk factors that have been shown to be associated with adolescents’ emotion regulation difficulties and self-injurious behaviors were included; (2) representativeness: the selected risk factors reflect characteristics of the Chinese sociocultural context; (3) expansibility: the selected risk factors are closely aligned with the developmental characteristics of early adolescents; (4) uniqueness: there are no inclusion or overlap relationships between the selected risk factors.

#### 2.3.3. Difficulties in Emotion Regulation Scale

The Difficulties in Emotion Regulation Scale (DERS) developed by [13] ([13]) was used in this study. The Chinese version was revised and validated by [41] ([41]) among Chinese adolescents and has demonstrated good reliability and validity. It consists of 36 items on six dimensions: emotional response acceptance, goal-directed behavior, impulse control, emotional awareness, emotional clarity, and the use of regulation strategies. A 5-point Likert scale was used (1 = almost never, 5 = almost always), with higher scores indicating greater difficulties in emotion regulation. In the present study, the construct validity of the scale was supported by confirmatory factor analysis (CFA), indicating an acceptable fit to the data (χ^2^(584) = 1576.33, CFI = 0.91, TLI = 0.93, RMSEA = 0.07, SRMR = 0.06). The internal consistency reliability was satisfactory, with a Cronbach’s α of 0.89.

### 2.4. Data Analysis

Tests of common method bias, descriptive statistics, and correlation analyses were performed using SPSS 27.0. LCA and LPA were conducted using Mplus 8.3. We also utilized Mplus 8.3 to further examine the relationships between the identified latent classes, latent profiles, and adolescent NSSI through logistic regression and BCH method (Bolck, Croon, and Hagenaars method).

LCA and LPA were conducted using Mplus 8.3, and models with 2–5 classes were compared. Model evaluation criteria included the Akaike Information Criterion (AIC), Bayesian Information Criterion (BIC), sample size-adjusted BIC (aBIC), entropy, the Lo–Mendell–Rubin adjusted likelihood ratio test (LMR-LRT), and the bootstrap likelihood ratio test (BLRT). Lower AIC, BIC, and aBIC values indicate better model fit. Entropy values closer to 1 reflect higher classification accuracy, with values above 0.8 suggesting over 90% accuracy ([27]). Significant *p*-values for the LMR-LRT or BLRT indicate that a model with k classes fits significantly better than a model with k-1 classes ([33]).

## 3. Results

### 3.1. Test of Common Method Bias

The Harman single-factor test identified 22 factors with eigenvalues greater than 1. The first factor accounted for 17.32% of the variance, below the critical threshold of 40% ([32]). There was no serious common method bias in this study.

### 3.2. Descriptive Statistics and Correlations

Table 1 shows that cumulative family risk and NSSI were significantly positively correlated, and both were significantly positively correlated with emotion regulation difficulties (total score and all dimension scores). Among the dimensions of emotion regulation difficulties, all dimensions, except for emotional awareness, demonstrated significant positive correlations with each other.

### 3.3. Latent Classes of Cumulative Family Fisk

A latent class model was established, and models with two to five classes were compared. The fit indices are presented in Table 2. Entropy values for the 2-class, 4-class, and 5-class models all exceeded 0.80, indicating satisfactory accuracy. Starting from the 2-class model, the BIC and aBIC progressively increased with the addition of more classes, indicating a decline in model fit. Furthermore, from the 2-class model onward, the *p*-values for LMR-LRT and BLRT were no longer significant. The 2-class model was chosen as the final model for this study.

Table 3 and Figure 1 provide the information used to characterize the classes in the 2-class model. The conditional probabilities (probability of being at risk) for each family risk factor were significantly higher in the C1 group compared to the C2 group, with particularly notable differences in the dimensions of family conflict (χ^2^ = 134.80, *p* < 0.001), father–child closeness (χ^2^ = 476.33, *p* < 0.001), and mother–child closeness (χ^2^ = 923.84, *p* < 0.001). The C1 group was labeled the “high-risk group” (30.78%) and the C2 group was labeled the “low-risk group” (69.22%).

As shown in Table 4, the BCH method was conducted. The self-injury level in the high-risk group (M = 6.33) was significantly higher than that in the low-risk group (M = 1.33).

### 3.4. Latent Profiles of Emotion Regulation Difficulties

A latent profile model was established, and models with two to five profiles were compared. The model fit information is shown in Table 5. As the number of classes increased, the AIC, BIC, and aBIC indices gradually decreased, indicating improved model fit. The entropy values for both the 2-class and 3-class models reached 0.80, indicating satisfactory accuracy. The LMR-LRT and BLRT values for the 3-class model were significant, indicating the 3-class model provided a better fit than the 2-class model. The 3-class model was selected as the optimal model for this study.

As shown in Table 6 and Figure 2, within the 3-class model, the mean values of Group C3 were the highest in all dimensions except emotional awareness, followed by Group C2 and Group C1. Particularly notable differences among the three groups in the use of regulation strategies were observed (F = 1981.96, *p* < 0.001). Group C1 was labeled as the “low-difficulty group”, Group C2 as the “medium-difficulty group”, and Group C3 as the “high-difficulty group”.

As shown in Table 7, the BCH method was conducted. The self-injury level in the high-difficulty group (M = 13.22) was significantly higher than in the other two groups. Furthermore, the self-injury level was significantly higher in the medium-difficulty group (M = 3.06) than in the low-difficulty group (M = 0.65).

### 3.5. Logistic Regression with Identified Latent Classes as Predictors of Identified Latent Profiles

As shown in Table 8, the latent class of cumulative family risk served as the independent variable (the low-risk group as the reference group, coded as 0), and the latent profile of emotion regulation difficulties was the dependent variable. When the low-difficulty group was the reference group, participants in the high-risk group were more likely to be classified into the medium-difficulty group (OR = 1.72, *p* < 0.05) or the high-difficulty group (OR = 5.63, *p* < 0.01). When the medium-difficulty group was the reference group, participants in the high-risk group were more likely to be classified into the high-difficulty group (OR = 3.27, *p* < 0.01).

## 4. Discussion

This study employed a person-centered approach, using LCA and LPA to investigate latent patterns of cumulative family risk and emotion regulation difficulties among adolescents, as well as their association with NSSI.

### 4.1. Latent Classes of Cumulative Family Risk and Their Association with NSSI

The LCA revealed that adolescents’ cumulative family risk could be categorized into two groups: high-risk group (30.78%) and low-risk group (69.22%). The probabilities of various risks were significantly higher in the high-risk group compared to the low-risk group, with particularly significant differences in the dimensions of family conflict, father–child closeness, and mother–child closeness. This suggests that the family atmosphere is a core component of cumulative family risk.

Adolescents in the high-risk group had significantly higher NSSI than those in the low-risk group, indicating that higher levels of cumulative family risk are associated with increased NSSI ([31]; [39]; [23]). The Systemic Model of Self-Injury suggests that a negative family atmosphere, characterized by frequent conflict and poor parent–child relationships, may be a key influence on adolescent self-injury ([6]). Family conflict undermines emotional regulation and significantly increases emotional insecurity and depressive symptoms, raising the risk of NSSI among adolescents ([15]). The parent–child relationship provides adolescents with emotional support and guidance ([19]). Without a close parent–child relationship, the adolescent might use NSSI as a substitute emotion regulation strategy ([37]). In summary, cumulative family risk, particularly risk due to a negative family atmosphere, is associated with a significant increase in the likelihood of adolescent NSSI.

### 4.2. Latent Profiles of Emotion Regulation Difficulties and Their Association with NSSI

The LPA revealed that adolescents’ emotion regulation difficulties could be categorized into three groups: low-difficulty group (56.02%), medium-difficulty group (32.60%), and high-difficulty group (11.38%). The differences among the three groups were significant in all dimensions except emotional awareness, with particularly notable differences in the use of regulation strategies.

The Emotion Dysregulation Model indicates that a lack of effective emotion regulation strategies is a key characteristic of individuals who engage in NSSI ([14]). Maladaptive strategies, such as expressive suppression, not only fail to regulate emotions effectively but may also intensify emotional distress, leading to the accumulation of negative emotions and resulting in NSSI ([34]). Interestingly, emotional awareness difficulties do not appear to be an important part of emotion regulation difficulties. Emotional awareness primarily entails recognizing the presence of emotions ([13]). [28] ([28]) found that the association between emotional awareness and emotional problems was more dependent on how individuals managed their emotions. Recent studies have highlighted the roles of emotional granularity and emotional acceptance as more central components of successful emotion regulation processes. Emotional granularity refers to the ability to precisely differentiate between emotional states, which facilitates more targeted and adaptive regulation strategies ([18]). Emotional acceptance involves a nonjudgmental attitude toward emotional experiences, allowing individuals to reduce secondary distress and respond more flexibly to emotional challenges ([20]). Future research may benefit from explicitly incorporating these dimensions to better understand the emotion regulation profiles of adolescents.

Adolescents in the high-difficulty group had the highest self-injury scores, followed by those in the medium-difficulty group and the low-difficulty group, indicating that adolescents with greater emotion regulation difficulties also exhibit higher levels of NSSI ([8]). The emotion regulation ability is a critical internal factor influencing the occurrence of NSSI ([25]). Compared to non-self-injurers, self-injurers often demonstrate lower emotional recognition, less acceptance of emotional responses, more impulsive behaviors, less use of adaptive regulation strategies, and greater difficulty maintaining goal-directed behavior ([45]; [24]; [8]). In conclusion, emotion regulation difficulties, particularly the absence of effective strategies, appear to contribute to adolescent NSSI.

### 4.3. Prediction of Emotion Regulation Difficulties Profiles by Cumulative Family Risk Classes

Adolescents in the high-risk group were more likely to be classified into the high-difficulty group, followed by the medium- and low-difficulty groups, compared to those in the low-risk group. These findings align with previous research, indicating that the family environment plays a significant role in shaping adolescents’ emotion regulation abilities ([29]) and that family risk factors are key contributors to emotion regulation difficulties ([12]; [15]). Without a supportive family environment, NSSI may serve as a maladaptive alternative for regulating emotions ([48]; [17]). In conclusion, when family risk leads to emotion regulation difficulties, NSSI may also increase.

### 4.4. Implications

From a theoretical perspective, this study contributes to the theoretical understanding of adolescent NSSI by employing a person-centered approach, offering a distinct perspective from traditional variable-centered methods. Through LCA and LPA, we identified subgroups of adolescents exhibiting different patterns of cumulative family risk and emotion regulation difficulties. The findings revealed that adolescents with higher levels of cumulative family risk were more likely to experience greater emotion regulation difficulties, which in turn were associated with higher levels of NSSI. These results highlight the critical roles of family risk and emotion regulation in the development of NSSI, aligning with key assumptions of the Systemic Model of Self-Injury ([6]) and the Affective Cascade Model ([34]).

From a practical standpoint, the findings suggest that intervention efforts should target both the environmental and emotional factors associated with adolescent NSSI. Schools could play a central role by implementing social–emotional learning programs that enhance students’ abilities in emotion recognition, expression, and regulation ([35]). In addition, schools should facilitate parents understanding of adolescent psychological development and impart skills for maintaining harmonious interactions through initiatives such as parent schools ([15]). Furthermore, parent-focused interventions, including workshops on nonviolent communication and responsive parenting, may help reduce cumulative family risk and foster healthier parent–child relationships ([22]). However, the implementation of these supports may face significant challenges in rural or under-resourced areas. To address these issues, low-cost, scalable interventions such as mobile-based psychoeducational programs and self-guided online courses should be considered ([16]). We believe leveraging existing school infrastructures and digital technologies could help bridge the service gap and extend emotional support to adolescents in need.

### 4.5. Limitations and Further Research

First, this study relied on self-reported measures. Future research could employ multi-source reporting to enhance result reliability. Second, cumulative family risk includes a broad range of factors. Beyond those examined in this study, future research could investigate the influence of additional family interpersonal factors (e.g., parental mental health) on adolescent NSSI. Third, this study used a cross-sectional design, offering only the patterns of adolescents’ current emotion regulation difficulties. Future studies could utilize longitudinal designs (e.g., latent transition analysis) to explore the dynamic trends of emotion regulation ability throughout adolescence.

## 5. Conclusions

(1)Adolescents can be classified into two latent classes of cumulative family risk: high-risk (30.78%) and low-risk (69.22%). Adolescents in the high-risk group had significantly higher self-injury scores than those in the low-risk group.(2)Adolescents exhibit three latent profiles of emotion regulation difficulties: low-difficulty (56.02%), medium-difficulty (32.60%), and high-difficulty (11.38%). Adolescents in the high-difficulty group had the highest self-injury scores, followed by those in the medium-difficulty group and the low-difficulty group.(3)Cumulative family risk latent classes predict emotion regulation difficulty latent profiles. Adolescents in the high-risk group were more likely to be classified into the high-difficulty group, followed by the medium- and low-difficulty groups, compared to those in the low-risk group.

## Figures and Tables

**Figure 1 behavsci-15-00543-f001:**
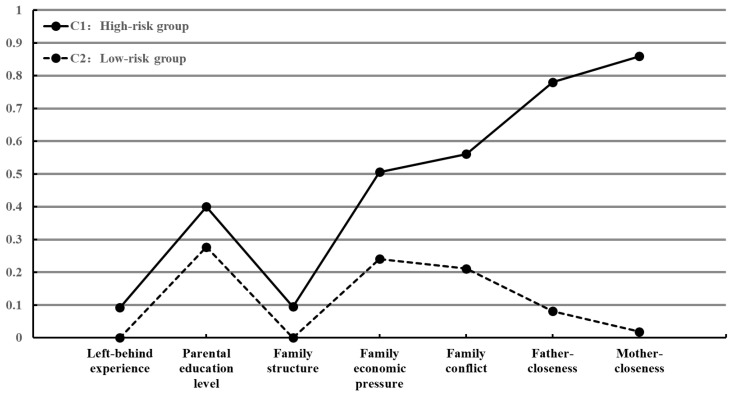
The conditional probabilities of each family risk factor across family risk groups.

**Figure 2 behavsci-15-00543-f002:**
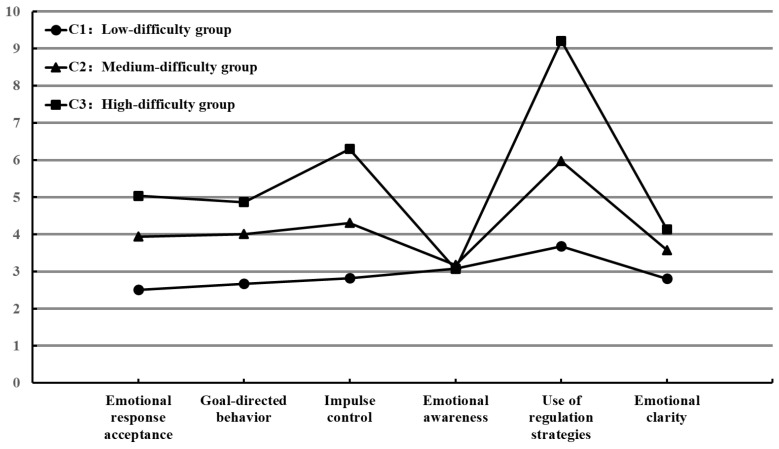
Differences in the means across emotion regulation difficulties groups.

**Table 1 behavsci-15-00543-t001:** The mean (M), standard deviation (SD), and correlations of the main variables.

	1	2	3	4	5	6	7	8	9
1. Emotional response acceptance	1								
2. Goal-directed behavior	0.44 ***	1							
3. Impulse control	0.55 ***	0.54 ***	1						
4. Emotional awareness	−0.11 ***	0.03	0.05	1					
5. Use of regulation strategies	0.62 ***	0.62 ***	0.71 ***	0.04	1				
6. Emotional clarity	0.34 ***	0.29 ***	0.37 ***	0.41 ***	0.43 ***	1			
7. NSSI	0.31 ***	0.25 ***	0.40 ***	0.07 *	0.48 ***	0.25 ***	1		
8. Emotion regulation difficulties	0.70 ***	0.72 ***	0.80 ***	0.33 ***	0.87 ***	0.65 ***	0.44 ***	1	
9. Cumulative family risk index	0.18 ***	0.23 ***	0.25 ***	0.13 ***	0.30 ***	0.22 ***	0.32 ***	0.32 ***	1
M	12.45	12.28	12.27	16.28	16.09	10.94	2.89	80.32	1.6
SD	5.15	4.71	5.05	5.24	6.58	3.78	7.49	20.92	1.47

Note: * means *p* < 0.05, *** means *p* < 0.001.

**Table 2 behavsci-15-00543-t002:** Fit indices for models derived from the LCA of cumulative family risk factors.

Model	AIC	BIC	aBIC	Entropy	LMR-LRT(*p*)	BLRT(*p*)	Size (% of Sample)
1	7021.98	7056.65	7034.41	-	-	-	-
2	6359.80	6434.09	6386.45	0.83	<0.001	<0.001	30.78/69.22
3	6357.77	6471.68	6398.63	0.74	0.063	0.097	8.41/23.52/68.07
4	6361.80	6515.33	6416.87	0.80	0.094	0.375	26.48/4.21/68.07/1.24
5	6368.76	6561.91	6438.04	0.84	0.052	0.973	0.96/5.64/24.47/67.11/1.82

**Table 3 behavsci-15-00543-t003:** Chi-square tests of conditional probabilities across family risk groups.

	High-Risk Group		Low-Risk Group		χ^2^ (df = 1)
	At Risk(1)	No Risk(0)	At Risk(1)	No Risk(0)	
Left-behind experience	0.09	0.91	0	1	69.45 ***
Parental education level	0.4	0.6	0.28	0.72	16.77 ***
Family structure	0.09	0.91	0	1	71.83 ***
Family economic pressure	0.51	0.49	0.24	0.76	68.06 ***
Family conflict	0.56	0.44	0.21	0.79	134.80 ***
Father-child closeness	0.78	0.22	0.08	0.92	476.33 ***
Mother-child closeness	0.86	0.14	0.02	0.98	923.84 ***

Note: *** means *p* < 0.001.

**Table 4 behavsci-15-00543-t004:** Differences in NSSI across cumulative family risk groups.

	High-Risk Group(M ± SE)	Low-Risk Group(M ± SE)	χ^2^ (df = 1)
NSSI	6.33(0.65)	1.33(0.19)	50.91 ***

Note: *** means *p* < 0.001.

**Table 5 behavsci-15-00543-t005:** Fit indices for models derived from the LPA of emotion regulation difficulties.

Model	AIC	BIC	aBIC	Entropy	LMR-LRT(*p*)	BLRT(*p*)	Size (% of Sample)
1	38,076.13	38,135.57	38,097.45	-	-	-	-
2	36,425.62	36,519.73	36,459.38	0.88	<0.001	<0.001	70.08/29.92
3	35,940.39	36,069.17	35,986.59	0.86	<0.001	<0.001	56.02/32.60/11.38
4	35,747.86	35,911.30	35,806.49	0.78	0.232	<0.001	27.44/8.70/40.34/23.52
5	35,627.54	35,825.65	35,698.61	0.75	0.202	<0.001	26.10/20.08/25.14/21.13/7.55

**Table 6 behavsci-15-00543-t006:** Analyses of variance of means across emotion regulation difficulties groups.

	C1(M ± SE)	C2(M ± SE)	C3(M ± SE)	F	Comparison
Emotional response acceptance	2.51(0.07)	3.94(0.16)	5.04(0.22)	462.88 ***	C3 > C2 > C1
Goal-directed behavior	2.67(0.07)	4.01(0.12)	4.87(0.17)	360.84 ***	C3 > C2 > C1
Impulse control	2.82(0.10)	4.31(0.22)	6.30(0.38)	711.27 ***	C3 > C2 > C1
Emotional awareness	3.08(0.10)	3.18(0.12)	3.07(0.13)	0.55	No significant differences
Use of regulation strategies	3.68(0.12)	5.97(0.23)	9.21(0.31)	1981.96 ***	C3 > C2 > C1
Emotional clarity	2.81(0.09)	3.58(0.14)	4.14(0.18)	120.32 ***	C3 > C2 > C1

Note: C1 = low-difficulty group, C2 = medium-difficulty group, C3 = high-difficulty group, *** means *p* < 0.001.

**Table 7 behavsci-15-00543-t007:** Differences in NSSI across emotion regulation difficulties groups.

	M ± SE	χ^2^ (df = 2)	
low-difficulty group	0.65(0.13)	low-difficulty group vs. medium-difficulty group	29.93 ***
		low-difficulty group vs. high-difficulty group	76.63 ***
medium-difficulty group	3.06(0.40)	medium-difficulty group vs. high-difficulty group	44.02 ***
high-difficulty group	13.22(1.43)	-	-

Note: *** means *p* < 0.001.

**Table 8 behavsci-15-00543-t008:** The prediction of cumulative family risk latent classes on emotion regulation difficulties latent profiles.

	REF: Low-Difficulty Group				REF: Medium-Difficulty Group	
	Medium-Difficulty Group		High-Difficulty Group		High-Difficulty Group	
High-risk group (coded as 1)	OR	CI(95%)	OR	CI(95%)	OR	CI(95%)
	1.72 *	[1.12, 2.64]	5.63 **	[3.52, 9.01]	3.27 **	[1.94, 5.50]

Note: * means *p* < 0.05, ** means *p* < 0.01.

## Data Availability

The raw data supporting the conclusions of this article will be made available by the authors on request.

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
