# Peer review of "Relationships Between Cumulative Family Risk, Emotion Regulation Difficulties, and Non-Suicidal Self-Injury in Adolescents: A Person-Centered Analysis"

_behavsci, 2025, doi:10.3390/bs15040543_

Round 1
Reviewer 1 Report
Comments and Suggestions for Authors
Thank you for the opportunity to review behavsci-3505676: The MS addresses an important issue that youth face. The authors used a sophisticated person-centered approach, and latent class and profile analyses, to investigate the interrelation between cumulative family risk, emotion regulation difficulties, and NSSI behaviors in a large sample of Chinese youth.
Here are some comments that may improve this manuscript:
The Systemic model of Self-Injury is mentioned, but it would be beneficial to elaborate on how family conflict and parent-child closeness interact with emotional dysregulation in daily life for these youth. What pathways or mechanisms might exist?
The discussion on emotional awareness being non-significant is intriguing. This could be further explored with reference to recent studies on emotional granularity versus emotional acceptance in the emotion regulation literature.
The practical implications were noted only in brief (the role of schools and family education), and the authors are advised to expand on the following: What types of school-based or parent-focused interventions might be most effective for adolescents identified in these groups? What barriers may hinder the implementation of these supports in rural or under-resourced areas, especially considering the context of "left behind" children?
Minor Editorial Suggestions: Clarify acronyms (e.g., LMR-LRT) at first use for interdisciplinary readers.
Comments on the Quality of English LanguageThis is a good ms. After minor calification, it can be accpeted for publication.
Author Response
Thank you for your comments concerning our manuscript. These comments are very valuable and helpful for the revisions and improvement of our manuscript. We have made revised according to the suggestion. Please see the attachment::

Reviewer 2 Report
Comments and Suggestions for Authors
This research is innovative, the full article is complete, and the data analysis is methodologically rigorous. Nevertheless, we put forward several recommendations:
- It is recommended that the abstract be organized into four sections: Background, Methods, Results and Conclusions.
- The abstract should briefly describe the statistical analysis methods used in this research.
-
What is the significance of examining the relationship between cumulative family risk, emotional regulation difficulties, and self-injury in early adolescence? It is written in the text that national research on the relationship between family risk and self-injurious behavior has focused on middle and high school students, so what is the purpose of this study to use early adolescents as the research subjects?
- Part 1.2 of the introduction suddenly writes about the difficulty of emotion regulation, but there is no mention of emotion regulation above, so it seems a bit abrupt to suddenly start writing about emotion regulation here, and it is suggested that the first part of the introduction briefly mention the content of emotion regulation.
- It is recommended that the research hypothesis be added at the end of the introduction.
- The data analysis section of the research methods is too simple and it is recommended that this section be described in detail.
- Implications of this study for adolescent psychosocial development and rehabilitative treatment can be added in the conclusion section.
Author Response
Thank you for your comments concerning our manuscript. These comments are very valuable and helpful for the revisions and improvement of our manuscript. We have made revised according to the suggestion. Please see the attachment:
Reviewer 3 Report
Comments and Suggestions for Authors
There are a number of issues I would like the authors to address:
COMMENT 1: The introductory section is well written and adequately referenced, although there are additional recent articles (past 3-4 years) that could be added to provide a more contemporary perspective on the research field.
COMMENT 2: The research is more than minimal risk research. As such I would like the authors to more clearly indicate what efforts were made to prevent research-related distress among participants and what resources were made available to participants in the event of such distress.
COMMENT 3: In cases where validated assessment instruments were used in data collection, please provide details regarding the validation/adaptation process employed in the development and/or translation of such instruments
COMMENT 4: Study implications are addressed largely in relation to the school's role in primary prevention. I believe that the manuscript would be enhanced if the authors more comprehensively addressed the implications of study findings for theory, practice, and future research
Author Response

(The authors gave the same response as above.)
